# Sources of diagnostic delay for people with Crohn's disease and ulcerative colitis: Qualitative research study

**AWARE-IBD Diagnostic Delay Working Group**[1,2,3,4]¶*

1 Sheffield CTRU, University of Sheffield, Regent Court, Sheffield, United Kingdom, 2 The Medical School, The University of Sheffield, Sheffield, United Kingdom, 3 Academic Unit of Medical Education, The Medical School, The University of Sheffield, Sheffield, United Kingdom, 4 Sheffield Inflammatory Bowel Disease Centre, Royal Hallamshire Hospital, Sheffield Teaching Hospitals NHS Foundation Trust, Sheffield, United Kingdom

¶ Membership of the AWARE-IBD Diagnostic Delay Working Group is provided in the Acknowledgments.
* d.hind@sheffield.ac.uk

**Data Availability Statement:** In line with the Standard Operating Procedures in place at The University of Sheffield, where this study was

## Abstract

### Objective

An improved understanding of the causes and experience of diagnostic delay in Inflammatory Bowel Disease (IBD).

### Methods

Framework analysis of semi-structured interviews with 20 adults with IBD.

### Results

Participants' prior knowledge of normal bowel function/IBD was limited. Symptoms were sometimes misattributed to mild/transient conditions or normalised until intolerable. Family pressures, work, education, mistrust of doctors, fear and embarrassment could exacerbate delays. Poor availability of face-to-face appointments deterred people from seeing a GP. Patients feared that by the time they got to see their GP, their symptoms would have resolved. Patients instead self-managed symptoms, but often regretted not seeking help earlier. Limited time in consultations, language barriers, embarrassment, and delays in test results subsequently delayed specialist referrals. GPs misattributed symptoms to other conditions due to atypical or non-specific presentations, leading to reduced trust in health systems. Patients complained of poor communication, delays in accessing test results, appointments, and onward referrals—all associated with clinical deterioration. GPs were sometimes unable to 'fast-track' patients into specialist care. Consultations and endoscopies were often difficult experiences for patients, especially for non-English speakers who are also less likely to receive information on mental health support and the practicalities of living with IBD.

conducted, data are archived at a dedicated location within the University's network. We have not shared de-identified interview transcripts, as this is concordant with University of Sheffield processes for trial data sets. Furthermore, our participant consent forms and information sheets (which have been approved by Health and Care Research Support Centre, Wales Research Ethics Committee 3) stated that the transcripts would be stored securely for 10 years at the University of Sheffield, and only shared in an anonymised form if considered appropriate by the research team. Therefore, uploading interviews transcripts onto a publicly accessed data repository would oppose what participants agreed to in the informed consent procedure. Data requests may be sent to ctru@sheffield.ac.uk.

**Funding:** This research was funded by Crohn's & Colitis UK. The funders had no role in study design, data collection and analysis, decision to publish, or preparation of the manuscript.

**Competing interests:** The authors have declared that no competing interests exist.

## Conclusions

The framework analysis demonstrates delay in the diagnosis of IBD at each stage of the patient journey.

## Recommendations

Greater awareness of IBD amongst the general population would facilitate presentation to healthcare services through symptom recognition by individuals and community advice. Greater awareness in primary care would help ensure IBD is included in differential diagnosis. In secondary care, greater attention to the wider needs of patients is needed–beyond diagnosis and treatment. All clinicians should consider atypical presentations and the fluctuating nature of IBD. Diagnostic overshadowing is a significant risk–where other diagnoses are already in play the risk of delay is considerable.

## Introduction

Inflammatory Bowel Disease (IBD) is becoming more common globally, with an incidence of 24.3 and 12.7 per 100,000 person-years in Europe for ulcerative colitis ("UC") and Crohn's Disease ("CD") respectively [1]. IBD is associated with substantial humanistic and economic burden which increase with symptom severity [2–4]. Median time to diagnosis from initial symptoms ranges from 2 to 84 months in CD and 2 to 114 (centring on 4) months in UC [5,6] with a median of medians of 8 and 4 months respectively. Delays in presentation, referral, and diagnosis are associated with poor clinical outcomes and quality of life, more emergency admissions, more hospital admissions in general, higher corticosteroid use, and greater health system and societal costs [7–9].

People with IBD often report symptoms for years before they are formally diagnosed [10], during which high levels of health care usage are reported [11]. 'Prodromes'–early signs or symptoms that indicate disease onset before diagnostically specific indicators–are frequent in CD and occur in half of UC patients [12], including not only physical but also mental health symptoms [13,14]. Initial symptoms are often intermittent, unnoticed or insufficiently troubling to trigger referral to a specialist. GPs may be unaware of different presentations and may not communicate developments which should cause patients concern [15].

People who report severe pain and diarrhoea may have minimal laboratory markers of disease activity [16]. Up to one third of IBD patients are initially assessed as having Irritable Bowel Syndrome (IBS) [12,17–20] Such a label doubles the duration of the pathway to diagnosis from an average of two to four years in Crohn's Disease [12,20–22]. In one third of these patients, the IBS label persists for five or more years [23]. The cumulative risk of a diagnosis of IBD after an initial diagnosis of IBS increases with the length of follow-up period [24]. In a 2019–20 benchmarking exercise, the wait from reporting symptoms to a GP to diagnosis was over 12 months in 26% cases; over a third of services self-assessed as unsatisfactory in terms of waiting times for pre-diagnosis endoscopy and histology, and this correlated closely with patient perception of overall service quality [25].

There has been no reduction in diagnostic delay over time [26]. In Crohn's Disease, diagnostic delay is greater [8] possibly because symptoms at presentation–abdominal pain, altered bowel habit, constipation or diarrhoea–are mistaken for other gastrointestinal disorders [12]. Recent systematic reviews have identified the need for further research into the reasons for

diagnostic delay, especially focusing on knowledge of IBD [5,6]. The objective of this study was an analysis of qualitative interviews exploring patient insight into the journey to a diagnosis of IBD, the barriers to gaining a diagnosis and the impact on their quality of life. For the purpose of this study, our funder defined diagnostic delay to have occurred when a person's diagnosis by a secondary care specialist took longer than 12 months from first onset of symptoms.

## Methods

### Methodological orientation and theory

We used a case study methodology [27], based on a synthetic model of previous theories, models and frameworks about barriers to timely presentation and healthcare (Fig 1). We used

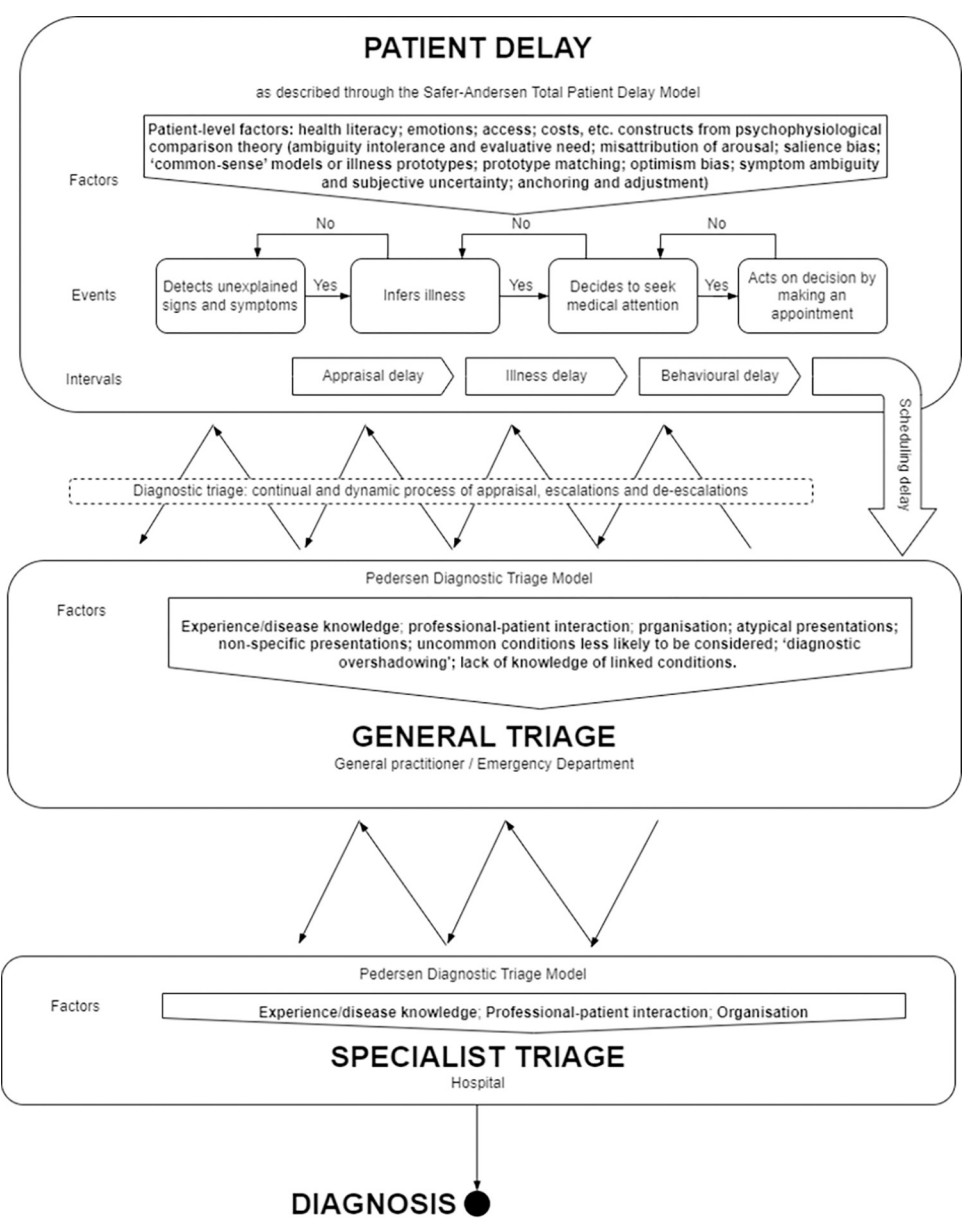

**Fig 1. Synthetic model underlying interview guide and analysis.**

**Table 1. Psychophysiological comparison theory.**

| Construct | Elaboration |
|---|---|
| Ambiguity intolerance and evaluative need | People are motivated to understand personally relevant symptoms of undiagnosed diseases, because uncertainty is discomforting |
| Misattribution of arousal | People are often not accurate in determining the causes of symptoms |
| Salience bias | People are more motivated to evaluate symptoms when symptom attributes are: (a) 'salient' (noticeable), unexpected or relevant to their prior health status; (b) 'consequential' in terms of being intolerable or disruptive. |
| 'Common-sense' models [32] or illness prototypes [33] | People develop 'common-sense' models, or prototypes of illness based on their own experience of health services and by social comparison with others. These prototypes may be activated by the appearance of symptoms. |
| Prototype matching | The more symptoms people can match to their 'common-sense' models, or illness prototypes, that they know about, the more likely they are to invoke that illness label. Conditions with low base rates are less likely to be selected as the cause of symptoms than more common disorders [32]. |
| Optimism bias | People tend to appraise symptoms as less-serious, transient and self-correcting, rather than life-threatening, chronic and requiring medical assistance, especially when symptoms are diffuse or confusing. |
| Symptom ambiguity and subjective uncertainty | When symptoms are more diffuse (e.g. fatigue) rather than specific and localised (e.g. rectal bleeding), there are a greater number of 'common-sense' models, or illness prototypes, to choose from, and there is a greater chance of misinterpreting the symptoms. |
| Anchoring and adjustment | Individuals are more likely to notice symptoms which are a good fit with their 'common-sense' models, or illness prototypes. People are selective in the way in which they monitor and test hypotheses about different events. As symptoms proliferate, worsen or improve, beliefs about the underlying illness change. |

Psychophysiological Comparison Theory (Table 1) to understand patient delay in appraising symptoms as requiring medical attention [28] and the Diagnostic Triage Model provides a descriptive framework to chart a circuitous path through generalist and specialist triage prior to diagnosis [29]. Psychophysiological Comparison Theory predicts that, when people notice symptoms, they try to understand their cause. They mentally compare the symptoms to their existing knowledge about illnesses. However, people are more likely to favour explanations that are less threatening. People can also describe their symptoms inaccurately because they have different perspectives and viewpoints when translating their subjective experiences into words. Social influences, especially wanting to meet certain expectations, can further change how people describe their symptoms. The Diagnostic Triage Model shows how people journey from first noticing symptoms to getting a diagnosis. There are decision points along the way– by you, your regular doctor, and specialist doctors. At each point, judgments are made–by people with symptoms, by generalists, by specialists, about how urgently the person need care. The model maps out the ups and downs on the path to diagnosis. Symptoms, people's instincts, who they talk to, and interactions with doctors all impact the speed of progress. Delays happen when concerns are minimised; faster diagnosis comes through appropriate worry and referrals. The model aims to explain why some diagnosis journeys are smooth and some frustratingly prolonged. We chose these theories because they have evidential and coherential virtues [30]: they have shown good fit with the empirical evidence across a range of settings; and, they posit factors that plausibly cause the effect (diagnostic delay) in question which are coordinated in an intuitively plausible whole. It is appropriate to combine two theories or models to overcome the inherent limitation of each [31]–Psychophysiological Comparison Theory focusing on the patient and the Diagnostic Treatment Model on the wider system.

## Participant sampling and approach

We conducted 20 interviews which, given the application of prior theory, was felt to be adequate to understand common experiences of delayed diagnosis, thereby achieving thematic saturation [34]. Patients who had been diagnosed in the previous three years, at least one year after symptom onset, were contacted through their preferred method of contact with a participant information sheet by a delegated researcher. We sampled for maximum variation from the South Yorkshire area of the UK, based on age at diagnosis, disease type, ethnicity and area-level deprivation using different methods: a researcher contacting participants, via their preferred method of contact on a hospital database of people with a consultant-confirmed diagnosis, who had already expressed an interest in research participation; REC-approved SMS messages from seven GP practices in areas of deprivation with high levels of non-English ethnicity and language competency; REC-approved emails to students attending The University of Sheffield, seeking people a diagnosis of IBD within three years, and other characteristics in which we were interested, such as ethnicity; clinician approach to emergency admissions.

## Conduct of interviews

To mitigate barriers to inclusion and participation, participants were given a choice of semi-structured interviews conducted by telephone or Google Meet (a secure video-communication service) between March and June 2022. Informed consent was obtained verbally before the interview. This was recorded onto an encrypted dictaphone and recorded onto a paper consent form, a copy of which was sent to the participant. Interviewees were offered a male or female interviewer. The interview guide (S1 File) was based on the synthetic model (Fig 1). Interviews were recorded on encrypted digital recorders and transcribed.

## Analysis

We undertook all stages of the National Centre for Social Research 'Framework' analysis approach within the latest version of NVivo (QSR International) [35]. Themes of *a priori* interest related to the synthetic model (Fig 1), with the exception that we did not systematically ask people how long it took from booking GP appointments to seeing a GP (scheduling delay). Using existing theories, models and frameworks (see above), thematic matrices and timelines, we matched the pattern of empirical data against the synthetic model and built explanations for why delays happen [36].

## Patient and public involvement (PPI)

We engaged diverse community members to ensure acceptability of study processes and as a form of member checking [37]. The study was designed in collaboration with two people with IBD who made extensive revisions to the interview guide before submission for approvals. While analysis was going on we circulated successive drafts of our findings to one PPI member and made amendments based on their comments. We convened three PPI workshops to ensure that we had incorporated sufficiently the views of three underserved populations: black and minority ethnic people, people from deprived Lower Super Output Areas (Deprivation Deciles 1 and 2), and people in the catchment areas of district general hospitals without specialist IBD care. A summary of findings was circulated to participants prior to each workshop. A Crohn's & Colitis UK Network Lead Coordinator chaired the workshops, during which team members presented findings and incorporated PPI member responses into our results sections, along with supporting points from the literature. Three PPI members amended the final draft of the report.

### Ethical approval

The study received ethical approval from the Health and Care Research Support Centre, Wales Research Ethics Committee 3 (16/08/2021). Informed consent was obtained verbally before all interviews. Consent was recorded onto an encrypted dictaphone and also recorded onto a paper consent form, a copy of which was sent to the participant.

## Results

Characteristics of the population are displayed in Table 2. Common prodromic symptoms are presented in Table 3. The sample included five people living in the most deprived, and one from the second-most deprived, deciles on the index of multiple deprivation. Abdominal pains (n = 16), bloating (n = 8) and/or cramps (n = 8), with increased frequency and looseness of bowel movements (n = 13) were often the first symptoms detected, as far back as ten years before diagnosis. Some participants reported long-term skin problems, such as puffiness, rashes, lesions, eczema or hives. Initially, symptoms were often short-lived with long intervals between flare-ups. Anaemia (n = 8) and fatigue (n = 14) were sometimes experienced later. Two participants suffered more from constipation, in one case with periods of with looseness. One participant had intermittent periods where their bowels were loose, but–unlike many– these did not escalate in frequency or intensity. All but two participants had symptoms that are considered high risk, individually or in combination.

### Patient delay

Interviews revealed examples of different aspects of the Patient Delay Model. One person acknowledged never having questioned what normal bowel movements were before. Three people with very long-term problems thought that their dysfunctional bowel movements were normal. One person said they knew that they had never had healthy bowel movements. Incorrect assumptions about the cause of symptoms (misattribution of arousal) included food intolerances (n = 5) and IBS (n = 7). Four participants reported using symptom diaries. Nine participants said that awareness of previously healthy bowel movements made emerging symptoms salient. Twelve participants described a slow escalation of symptoms, sometimes with change points where new symptoms were noted; four reported the sudden onset of very serious symptoms. Participants with slow onset symptoms reported the frequency, duration, and intensity of flare-ups increasing gradually in a relapsing-remitting pattern.

**Table 2. Population characteristics.**

| Characteristic | Cases |
|---|---|
| Female (n = 13) | P01, P02, P05, P07, P08, P09, P10, P11, P12, P13, P15, P18, P19 |
| Male (n = 7) | P03, P04, P06, P14, P16, P17, P20 |
| White British (n = 13) | P04, P05, P08, P09, P10, P11, P13, P14, P15, P16, P17, P18, P20 |
| White Other (n = 1) | P07 |
| British Asian (n = 5) | P01, P03, P06, P12, P19 |
| Arab (n = 1) | P02 |
| English not first language (n = 2) | P02, P07 |
| Interview conducted through interpreter (n = 1) | P02 |
| Crohn's Disease (n = 8) | P01, P08, P09, P10, P11, P13, P18, P19 |
| Ulcerative colitis (n = 8) | P02, P03, P04, P05, P14, P16, P17, P20 |
| Ulcerative proctitis (n = 1) | P12 |
| IBD unclassified (n = 2) | P07, P15 |

**Table 3. Prodromal symptoms or clinical features reported by patients, categorised as important alone or in combination by Atia et al. [38] and patient-estimated delay by category in months (m).**

| Case | Important individually | | | Important in combination | | | | | Category of Delay | | |
|------|-----------|-----------|-----------|-----------|---------|---------|---------|---------|---------|---------|---------|
| | Diarrhoea >1w | Bloody stool >1w | Elevated calprotectin | Abdominal. pain >1m | Anaemia | Chronic fatigue | Family history | Joint pain, uveitis, etc. | Patient delay (m) | General triage (m) | Specialist triage |
| Crohn's | | | | | | | | | | | |
| P01 | ✓ | - | ✓ | ✓ | ✓ | ✓ | - | ✓ | 114 | 4 | 2.5 |
| P08 | - | - | - | ✓ | - | ✓ | - | - | 0.5 | 4 | 60 |
| P09 | ✓ | - | ✓ | ✓ | - | ✓ | - | - | 2 | 8 | 10 |
| P10 | ✓ | - | ✓ | ✓ | ✓ | ✓ | ? | - | 120 | 6 | 18 |
| P11 | ✓ | - | - | ✓ | - | ✓ | ✓ | ✓ | 240 | 9.5 | 6 |
| P13 | ✓ | ✓ | ✓ | ✓ | - | ✓ | - | - | 6 | 4 | 3 |
| P18 | ✓ | - | - | ✓ | - | - | - | - | 192 | 24 | 2 |
| P19 | - | ✓ | - | ✓ | - | ✓ | ✓ | - | 48 | 13 | 11 |
| Ulcerative Colitis | | | | | | | | | | | |
| P02 | ✓ | - | - | ✓ | ✓ | ✓ | ✓ | - | 36 | 12 | 4 |
| P03 | ✓ | ✓ | - | ✓ | - | ✓ | - | - | 16 | 2 | 3 |
| P04 | ✓ | ✓ | - | ✓ | - | ✓ | - | - | 3.5 | 6 | 0 |
| P05 | - | ✓ | - | ✓ | - | ✓ | - | - | 1.5 | 4 | 9 |
| P06 | ✓ | ✓ | ✓ | ✓ | - | - | - | ✓ | 13 | 0.5 | 0.1 |
| P14 | ✓ | ✓ | - | ✓ | - | - | ? | - | 60 | 120 | 384 |
| P16 | ✓ | ✓ | - | - | - | - | - | - | 0 | 60 | 0.5 |
| P17 | - | - | - | ✓ | - | ✓ | - | - | 120 | 60 | 0.3 |
| P20 | - | ✓ | ✓ | - | - | - | - | - | 5 | 6 | 0 |
| Proctitis | | | | | | | | | | | |
| P12 | ✓ | ✓ | - | - | - | - | - | - | 2 | 6 | 6 |
| IBD-U | | | | | | | | | | | |
| P07 | ✓ | ✓ | - | ✓ | ✓ | ✓ | ✓ | - | 0 | 6 | 4.3 |
| P15 | - | ✓ | - | - | - | - | - | - | 3 | 24 | 1 |

Six participants said that the fact they did not understand their symptoms was stressful (ambiguity intolerance) and cited an urge for greater understanding (evaluative need). Symptoms were disruptive in terms of social isolation, strain on family relationships, loss of employment, and interruptions to the working day or education. Some talked about a point at which symptoms became intolerable.

Participants responses indicated that they had little opportunity to develop an IBD illness prototype through social comparison: nine had heard of IBD; four had some definite family history, others had relatives with lifelong symptoms consistent with IBD; prompted by conversations with others, six had considered the possibility of an IBD diagnosis (prototype matching) Six participants thought they might have bowel cancer. Others reported delayed help-seeking because they appraised symptoms as less-serious, transient and self-correcting, in line with available illness prototypes (food intolerances, IBS, infections), especially when symptoms were diffuse (joint pain, fatigue), coincided with another healthcare event/process health state (pregnancy, vaccination [39]). As symptoms multiplied or intensified two people changed their illness hypothesis through internet research (P01) and social comparison (P19).

Six participants reported using over-the-counter medications: paracetamol (n = 6); ibuprofen (n = 2); buscopan (n = 4); anusol (n = 1); imodium (n = 2); laxatives (n = 2); and probiotics (n = 1). Participants reported putting off help-seeking due to: a belief they could self-manage

**Table 4. Cases in which patient delay formed over 50% of the total delay period.**

| Cases | |
|---|---|
| Crohn's Disease | |
| P01 | Diffuse, intermittent symptoms, normalised over ten years until an acute flare in conjunction with an important life event forced her into help-seeking. |
| P11 | Older person whose childhood symptoms were misdiagnosed, and whose diagnostic labels 'stuck'. |
| P10, P18 | Symptoms arose later in life and attributed to menstrual / menopausal activity. |
| Ulcerative colitis | |
| P02, P03, P17 | Slowly escalating symptoms normalised, until symptoms became intolerable. |
| P06 | Diagnosed with uveitis by ophthalmologist, but no investigations for systemic disease. |

(n = 4) or that they could/should, cope (n = 3); trivialisation of symptoms (n = 3); a belief that the problem was self-limiting (n = 4); prioritising work (n = 3), university (n = 1) or childcare (n = 3); mistrust in doctors (n = 1); fear of severe diagnoses (n = 2); embarrassment about an intimate subject (n = 1), being seen to complain by family (n = 1) or burdening doctors (n = 1); and difficulty getting a face-to-face appointment with the GP (n = 3). Some patients reported delaying repeat help-seeking after GPs misattributed symptoms, leading to a different diagnosis.

People reported seeking help because symptoms had gradually become intolerable or suddenly become salient: pain (n = 11); bloating (n = 2); fatigue (n = 6); weight-loss (n = 2); blood in stool (n = 6); persistent vomiting (n = 2); or frequency or consistency of bowel movements (n = 8). Four were encouraged to seek help by family members; seven sought help because symptoms were interfering with caring activities or employment.

Individuals reported delays in help-seeking could be minimised or exacerbated by: response-control factors [40] (exacerbated by difficulties getting time off work, n = 3 or moving between home and university settings, n = 1); social norms (family pressure, n = 4); cognitive factors (active evaluation of arguments for/against their illness prototypes, n = 2); external factors (the need for symptom control before a job interview (n = 1) or to continue with caring/work activities (n = 7); anxiety at increasingly alarming symptoms (n = 3). At least four expressed regret at delaying help-seeking. Over 50% of the overall diagnostic delay period was broadly attributable to patients in seven cases (Table 4).

Sixteen participants first sought help from a GP; two went to accident and emergency, of which one was referred back to their GP after normal blood test results; two went to NHS walk-in centres, both then seeking a second opinion from their GP soon afterwards.

## General triage

Thirteen participants expressed comfort in discussing symptoms with the health professionals. Of the remaining seven participants, six expressed some discomfort in discussing symptoms during their consultations and one could not recall how they felt. Participants raised concerns around: addressing all the symptoms in a time-limited consultation (n = 1); difficulty describing symptoms, because of language barriers or the consultation being by telephone (n = 2); embarrassment about describing symptoms (n = 4, including diarrhoea and pain during sex), or being offered an intimate examination by a member of the opposite sex (n = 2). Eleven described initial encounters with GPs as supportive, feeling like they were being taken seriously and listened to. Five felt that the GP was insufficiently responsive to the severity of symptoms and/or that the GP's consideration of diagnostic possibilities was superficial or limited.

GP questions focused on the nature and duration of pain, changes in bowel movements, bleeding, stress at work, family history of the symptoms and of bowel cancer specifically. Four participants did not feel they were asked enough questions.

Participants reported that GPs requested blood (n = 14), stool (n = 9), and urine (n = 1) samples for analysis, in one case only after several months' of consultations. Four patients reported not being asked for any samples for testing, although three of these were referred to a consultant (i.e. a GP might assume relevant tests would be done at that stage). Where tests or endoscopies were ordered, participants reported feeling: positive because their symptoms were being taken seriously, or they felt they would get to the bottom of the problems (n = 6); concerned, anxious, worried, sad or scared (n = 8); angry, because they were having tests for a condition from which they had been 'un-diagnosed' (n = 1).

Participants reported waiting for less than one week (n = 1), one week (n = 4), two weeks (n = 2), a month or 'a few weeks (n = 2)', for blood and stool tests to be available (Qu 3–08). One participant reported waiting long, but unspecified, periods of time after submitting samples for testing. Two participants reported having to chase the GP surgery for test results. Four participants made comments to the effect that waiting time for tests was acceptable. Participants reported waiting for around 2 months (n = 1) and 3–4 months (n = 1), for endoscopies, with COVID-19 being cited as a cause of delay. One participant said they did not have to wait long. One participant felt that they were themselves partially responsible for delays in accessing endoscopies. One person voiced concern that with the cyclical nature of their symptoms, colonoscopies took place too late to document inflammation and this affected time to diagnosis. Others discussed delays in booking with worsening of symptoms while waiting for endoscopies. Ten participants clearly reported that GPs referred them to a gastroenterologist. One was referred to a dietician when diagnosed with IBS. Others reported that GPs did not refer them (n = 2). Three participants reported arranging private consultations for themselves, in one case outside the UK, back in their home country. Five participants self-presented at A&E before diagnosis. In two cases, due to symptom escalation or high test result scores, GPs referred patients directly to acute hospital wards. During the time participants were receiving primary care consultations, one described improved symptoms, one persistent, and ten described worsening symptoms.

Participants were diagnosed with other conditions before their IBD diagnosis, including: haemorrhoids (n = 2); IBS (n = 4); diverticulitis (n = 1). Patients reported GPs had difficulty recognising IBD as a source of their symptoms for a number of reasons (Table 5) [41].

It can be particularly easy to explain away symptoms if they can be attributed to pre-existing conditions or health states (such as pregnancy) [50]. Participants reported receiving, or having been recommended, the following treatments from/by healthcare professionals before diagnosis (Table 6): Five participants acknowledged declining tests or investigations (Table 7).

It is worth considering the likely causes of delay for those participants whose period of general triage was long in absolute terms (Table 8).

## Passage between general triage and patient delay / specialist triage

While our timelines depict the diagnostic pathway as a linear route through successive stages, the Pedersen model (Fig 1) predicts that there may be "dynamic movements. . . between. . . general triage and specialist triage" [29]. They conceptualise this as a "zig-zag line" illustrating how patients "can move back and forth between the different levels of triage before experiencing progress and obtaining the diagnosis". Several participants described such a path. P10 presented with peri-menstrual diarrhoea over 14 years before diagnosis. A colonoscopy did not show evidence of her to be IBD-negative at this stage and she tolerated symptoms under a

**Table 5. Reasons for difficulty in recognising IBD as source of symptoms.**

| | |
|---|---|
| Atypical presentations: Few or no prototypical features or unexpected test values, such that the correct diagnosis is either not generated or is rejected as not conforming to the clinician's disease prototype. | Six patients presented with constipation, when IBD is prototypically associated with stool looseness: "I didn't fit the boxes because of the constipation and my weight was stable so nobody thought about looking into Crohn's" (P11) |
| Non-specific presentations: symptoms that do not easily discriminate between different potential diagnoses (IBS, coeliac, dyspepsia, etc). | "He just kept basically diagnosing me with. . . stomach-ache" (P14) <br> "They were testing for coeliac. I cut all wheat and everything out (P10) |
| Uncommon conditions less likely to be considered [42]: Clinicians think only 'pathognomonic' indicators, those that are specific to a particular disease, will make a more uncommon disease more likely. Even when indicators (e.g. elevated faecal calprotectin) are present doctors may be cautious, because of the possibility of false positives. | In this example, a GP requested a second stool sample, which recorded insufficiently high calprotectin to detect Crohn's disease, which was finally diagnosed ten years later, and the GP defaulted to the more common diagnosis of IBS: "I needed two stool samples with high enough markers to be referred to gastro and [the second] wasn't high enough, and [I was told] that it was probably the IBS playing up and just to change my diet again." (P11). |
| 'Diagnostic overshadowing'[43]: where a patient has another condition which provides a credible explanation for symptoms or alters the presentation. It can be particularly easy to explain away symptoms if they can be attributed to pre-existing conditions or health states (such as pregnancy) [44]. | "They looked at that and thought, right the blood is from the haemorrhoids." (P17) <br> "Before I was 20, I'd got a diagnosis of hyper-mobility, fibromyalgia and IBS. . . and. . . a couple of mental health problems. . . I walk into the doctor's and everything is put down to them. . . they won't do any tests, they won't do anything. . . the past twenty years where I've been gas-lit and told everything was in my head." (P11, Crohn's disease) |
| Lack of knowledge of linked conditions. Uveitis is a rare autoimmune disease of the eye, that often occurs in combination with other systemic diseases, requiring collaborative work-up [45–47] between general practitioners, ophthalmologists, rheumatologists, neurologists and gastroenterologists [48], although this does not always happen. In isolation, it should trigger investigations for IBD [49]. | "I had the eye infection. . . And I went to the hospital, and they told me that I had uveitis. . . then [12 months later] I was losing a lot of blood. . . . I had, like, a really massive diarrhoea. . . every half an hour I was going to the bathroom and I was losing a lot of blood. . ." (P06, ulcerative colitis) |

diagnostic label of IBS for a further 10 years before seeking help again. P05 saw a consultant surgeon after three months of GP consultations concerning rectal bleeding, initially attributed to haemorrhoids. As she was nine months pregnant, he recommended a flexible sigmoidoscopy postpartum. She subsequently saw a GP again before that referral was made. P07's eligibility notes make it clear that they had presented at A&E with abdominal pain five times and been referred to a consultant gastroenterologist once, before the account in their interview starts with a GP visit.

**Table 6. Treatments recommended/received before diagnosis.**

| Test | Participants |
|---|---|
| Buscopan for pain (n = 2) | P01, P19 |
| Vitamin D and/or iron for anaemia (n = 3); | P02, P07, P10 |
| Oral or suppository laxatives for constipation (n = 5) | P03, P05, P11, P15, P20 |
| Steroid creams for haemorrhoids (n = 3); | P05, P08, P20 |
| Amitryptiline for looseness (n = 1); | P09 |
| Proton pump inhibitors for looseness (n = 1); | P09 |
| Synogut (beverin) for suspected IBS (n = 1); | P13 |
| Imodium for looseness (n = 1). | P18 |
| Pentasa (mesalazine) before a formal diagnosis, once IBD was suspected (n = 2) | P07, P10 |

**Table 7. Tests and investigations reported declined.**

| Test | Participant and annotations |
|---|---|
| Ultrasound scan (n = 1) | P02: at a time when symptoms were un-concerning |
| Intimate examination (n = 1) | P05: When offered by a male doctor. |
| Endoscopy (n = 1) | P14: After a previous bad experience;—an MRI was arranged instead. |
| Stool sample (n = 1) | P19 due to feeling uncomfortable about it |

## Specialist triage

Case studies of long specialist triage are presented in Table 9. Participants understood that their diagnosis was confirmed after: a colonoscopy (n = 16); an MRI (n = 1); barium contrast imaging (n = 1); pill camera (n = 1); blood tests (n = 1). Three participants reported difficult colonoscopy experiences, involving absent or inadequate pain control, or translation problems. Four discussed being told or overhearing their presumed diagnosis during an endoscopy. Consultants delivered the diagnosis: face-to-face (N = 14); by telephone (N = 2); by letter (N = 4). One participant received their diagnosis from a practitioner in their home country after long waits for consultations in the UK. Three people gave reports to the effect that they were dissatisfied with the quality of information-giving and/or did not receive written information on their condition. On diagnosis, people described feeling: *relieved* that their symptoms weren't 'all in their head', that they could receive treatment, or that they were not being diagnosed with bowel cancer; *sad*, because the diagnosis was of an incurable disease and the loss of the future life they had imagined; *concerned or anxious*, in general, about mortality, about their ability to care for dependent family members; *upset*, *distraught*, *scared* or *overwhelmed*. One participant was annoyed, having previously had a diagnosis of IBD retracted, and then reinstated after many years of symptoms.

**Table 8. Cases in which primary care delay formed the majority of total delay.**

| Cases | |
|---|---|
| Crohn's Disease | |
| P18 | P18's delay (two years in general triage, deprivation decile 2) can be partially attributed to atypical presentation, in the form of negative test results. |
| P19 | Non-specific presentations and relative disease prevalence (13 months in general triage, deprivation decile 6) misdiagnosed with IBS. |
| Ulcerative colitis | |
| P14 | 47-year-old person with UC, ten or more years in general triage, deprivation Decile 6) is an older patient who perceived their GP as out-of-touch and having trivialised their symptoms. Their experience could also be considered an example of misdiagnosis due to a non-specific presentation. "It was our local village GP, I think he'd come from the dark ages. . . some elderly man. . . I think he got his medical training in the 1940s and hadn't really kept himself particularly up to date with modern medicine. . . I kept being diagnosed with, like, you know, 'stomach aches'" (P14) |
| P16 | Both non-specific presentations and relative disease prevalence may have influenced the misdiagnosis of P16 (five years in general triage—over 50% of total diagnostic delay, deprivation decile 2), |
| P17 | Non-specific presentations and relative disease prevalence (five years in general triage, deprivation decile 1) misdiagnosis with IBS, |
| IBD unclassified | |
| P15 | Two years in general triage, deprivation Decile 5) where multiple alternative hypotheses (thyroid issues, haemorrhoids, diverticulitis) were pursued before IBD was picked up during routine bowel cancer screening. |

**Table 9. Case studies of long specialist triage.**

| Cases | |
|---|---|
| **Crohn's Disease** | |
| P08 | Under the care of another clinical directorate after diagnoses of haemorrhoids and polyps, brought them into an annual bowel cancer screening programme, through which their Crohn's disease was eventually diagnosed. |
| P09 | Received a negative endoscopy (atypical presentation) and had continuity of care disrupted by the to-and-fro between home and university. |
| P11 | Delays in access to endoscopy and MRI during COVID; also declined one test when offered. |
| P14 | Diagnosed, undiagnosed, re-diagnosed and re-diagnosed again over a period of decades. |
| **Ulcerative colitis** | |
| P05 | Presented to a specialist during pregnancy and, in the absence of a pre-arranged postpartum appointment, delayed re-presentation until symptoms escalated. |

Eleven people were satisfied with the way their diagnosis was conveyed; nine felt the diagnosis was not broached sensitively. Fourteen participants reported being immediately started on treatment, although in some cases, delays and errors were caused by the need for GPs to prescribe and for patients to collect from pharmacies. Delays of between two to eight weeks from diagnosis were experienced, waiting for further consultations, tests or instructions from the specialist IBD team. At the time of interview, five people reported that initial treatment controlled symptoms, although one was experiencing intolerable side effects at the time of interview; eight people said initial treatment failed to control symptoms; four reported that they were being considered for surgery or had already had it.

## Discussion

This qualitative study complements larger scale survey research [41,48,49] and illustrates that delay can occur at all stages of the pathway to diagnosis. The use of a synthetic model has allowed the points at which delay occurs to be mapped. It combines the Safer-Andersen total delay model to capture patient related causes of delay and the Pedersen diagnostic triage model of general and specialist triage for delays before and after reaching specialist care respectively. This has been supplemented by the use of psychophysiological comparison theory.

Patient -related delay was seen from the inability to develop an IBD prototype for themselves–for example, as a result of lack of awareness that their change in bowel habit was abnormal, from lack of knowledge of IBD as a potential explanation for their symptoms or from misattributing symptoms to another, less serious cause. Change in the diagnosis prototype was changed by individuals after research -particularly on-line. Delay was also caused when individuals prioritised other aspects of their lives over seeking help for their symptoms.

Interviewees represent a range of social backgrounds. Entry to the study cohort was specifically restricted to those with a confirmed diagnosis, including only those who were recently diagnosed to minimise recall bias and the presence of collective 'illness identifies' which arise with long exposure to advocacy groups [51]. We were unable to recruit frequently misdiagnosed perianal and proximal disease subgroups.

People with undiagnosed symptoms may not have heard of IBD (or CD or UC specifically), and delay seeking help because they misunderstand the cause of symptoms, minimise symptom severity, assume that doctors will do the same, or feel that symptoms will disappear by the time of consultation. People may normalise symptoms and prioritise work, study, or family commitments, over help-seeking. They are often hesitant to talk about bowel movements and may not understand what is abnormal and when to seek help.

Misattribution of presenting symptoms to more common, less serious health states (including IBS, pregnancy and menstruation) [44,52,53], without further investigations, can lead to premature closure of the diagnostic process [54]. GPs are less likely to investigate women with iron deficiency anaemia than men, as anaemia is common in menstruating women, although it is commonly also a presenting symptom of GI cancers, coeliac disease, and IBD, meaning there is potential to miss serious disease [55,56]. GPs may misattribute symptoms or to other previously diagnosed conditions, without further investigations, leading to premature closure of the diagnostic process [53]–a 'sticky' diagnosis (i.e. one that is difficult to shed) [57]–and patients disengaging from the health system. Premature closure of the diagnostic process delays symptom relief both for the 'good patient'[50,58,59], and for the 'tricky patient' who clinicians find confusing or threatening [60]. Community gastroenterology clinics may be associated with shorter referral and waiting times, enhanced continuity of care and improved onward referral to secondary care [54].

People often wait long periods for tests to be conducted, results to be obtained, appointments, and referrals, and do not want to be left 'in limbo' [55,56]. Unexplained symptoms [61,62] and poor communication [63] increase mistrust in, and complaints against, doctors.

Once the diagnosis is made, clinical specialists may deliver 'bad news' of a diagnosis without taking patient needs fully into account. These needs encompass the medium used to communicate as well as the content and breadth of communication, especially with regard to additional information regarding future developments and monitoring, as well as sources of further information and support. Colonoscopy is obviously central the process of diagnosis but was an area of concern. Maintaining comfort and pain relief is an important part of standard but in context of a potential diagnosis of IBD, how information is at the time of the procedure and follow-up arrangements should be carefully addressed in a unit.

## Recommendations

Education to raise awareness of IBD would help address the multiple stages contributing to delay. This may be useful in different forms for the wider public, schools and GPs, so that gastrointestinal symptoms are recognised as significant by individuals, friends, and family and in primary care. Using visual aids such as the Bristol stool scale to educate about normal bowel movements would benefit awareness of possible IBD symptoms and also those of other conditions [64]. Patients could benefit from guidance on questions to ask about their symptoms and the range of diagnostic possibilities (IBS, coeliac disease, bowel cancer as well as IBD), as well as encouragement to initiate follow-up consultations and second opinions, which can have a major impact on diagnosis outcomes [65]. Clinical initiatives are needed that combine transdiagnostic early warning scores (for cancer and these significant but 'benign' conditions) with clinical education and cross-system approaches to prevent premature closure of diagnosis [66,67]. From the experience described in this study, red-flag systems [39] would have triggered earlier referral, and could expedite the correct diagnosis of IBD patients inaccurately labelled as IBS [68], women with IBD whose anaemia and GI symptoms are attributed to menstruation [69], and other patients who are misdiagnosed with other conditions, for example haemorrhoids [70]. Computerised decision support systems which employ dynamic vocabulary tools could improve diagnostic accuracy [71].

Safety-netting is a practice designed to understand and communicate uncertainty, plan for follow-up, and improve the communication of interactions with laboratories and hospitals [72,73]. GPs should give and record tailored advice, in simple terms, whenever there is diagnostic uncertainty (e.g. from IBS or haemorrhoids to IBD), gastroenterologists should do likewise when there is potential for a change in diagnosis, using generic recommendations [74].

Clinicians should offer greater information on the implications of the working diagnosis, i.e. what to expect in terms of symptom change, how to self-care, and when to be concerned about changes in symptoms or their severity. Enabling and advising about greater patient self-advocacy would also be of benefit. Patients, GPs and gastroenterologists should co-produce trans-diagnostic red-flag systems, pathways and safety-netting systems for conditions characterised by overlapping GI and/or autoimmune symptoms. They should account for cognitive biases [75–80] and illness prototypes [42] by considering atypical IBD presentations for constipated, calprotectin-negative or borderline, or colonoscopy-negative patients.

Consultations and endoscopies can be difficult experiences, especially when patients do not speak English, or do not speak it well. Patient experience is a critical aspect of endoscopy quality [81] and frameworks already exist to inform improvement, especially in terms of pain relief, information-giving and the care relationship [82]. These frameworks should be more widely deployed.

Diagnosis can be a relief, but the implications and information can be overwhelming. Clear and compassionate communication and support are important at a distressing time. A toolkit for communicating IBD diagnoses should be co-produced, dealing with issues of understanding patients' preferred information needs and communication styles, the importance of contact medium (face-to-face, online, telephone, letter) and how to adapt and deploy different communication methods according to patient needs and preferences. Attention should be paid not just to the diagnosis communication but to wider issues such as future care implications, as well as to other sources of information and support. Greater attention should be paid to ensuring continuity of care, to improve clinician-patient communications and understanding.

The end-goal should be to both explain a course of action and build a relationship, given the importance of patient investment in the treatment process. Three quarters of patients want more information and support at diagnosis and half feel uncomfortable talking about psychosocial issues [83]. Consultants may benefit from consultation skills training to maintain focus on the patient during interpreted sessions, to break bad news sensitively [84,85], including via telemedicine [86], and to meet different informational and emotional needs during diagnosis [87]. Patient needs include information about the purpose of concomitant therapies (such as chemo-preventives), mental health support and the practical issues of living with IBD, including requesting reasonable adjustments from employers and concessions under disability legislation. People who are non-English-speaking are less likely to get this information and a toolkit for consultations covering "questions to ask your health professional" should be produced. Post-diagnosis delays and errors in dispensing should be addressed using local action plans, with form letters for urgent dispensing.

## Conclusions

Patients, generalists and specialists all contribute to delays in diagnosis of Crohn's disease and ulcerative colitis. Public education can play a role in encouraging earlier help-seeking for symptoms. Clinical education and awareness can be improved, especially in refining both the diagnosis process and experience, not least where other or prior diagnoses are in place. Further work is needed to re-engineer clinical systems and processes to reduce diagnostic delay.

## Supporting information

**S1 File. Interview topic guide.**
(DOCX)

## Acknowledgments

**AWARE-IBD Diagnostic Delay Working Group**

Daniel Hind c/o. Sheffield CTRU, University of Sheffield, Regent Court, 30 Regent Street, Sheffield, S1 4DA, UK.

Elena Sheldon c/o. Sheffield CTRU, University of Sheffield, Regent Court, 30 Regent Street, Sheffield, S1 4DA, UK.

George Lillington c/o. Sheffield CTRU, University of Sheffield, Regent Court, 30 Regent Street, Sheffield, S1 4DA, UK.

Nancy Greig c/o. Sheffield CTRU, University of Sheffield, Regent Court, 30 Regent Street, Sheffield, S1 4DA, UK.

Vicky Buckley c/o. Sheffield CTRU, University of Sheffield, Regent Court, 30 Regent Street, Sheffield, S1 4DA, UK.

Manfredi d'Afflitto c/o. Sheffield CTRU, University of Sheffield, Regent Court, 30 Regent Street, Sheffield, S1 4DA, UK.

Naseeb Ezaydi c/o. Sheffield CTRU, University of Sheffield, Regent Court, 30 Regent Street, Sheffield, S1 4DA, UK.

Basma Ali, The Medical School, The University of Sheffield, Beech Hill Road, Sheffield, S10 2RX, UK

Haifa Haamed, The Medical School, The University of Sheffield, Beech Hill Road, Sheffield, S10 2RX, UK

Tawfik Alhashemi, The Medical School, The University of Sheffield, Beech Hill Road, Sheffield, S10 2RX, UK, c/o/ Academic Unit of Medical Education,

Chris Burton, Academic Unit of Medical Education, The Medical School, The University of Sheffield, Beech Hill Road, Sheffield, S10 2RX, UK

Matthew Lee, c/o/ Academic Unit of Medical Education, The Medical School, The University of Sheffield, Beech Hill Road, Sheffield, S10 2RX, UK

Alan J. Lobo, Sheffield Inflammatory Bowel Disease Centre, Royal Hallamshire Hospital, Sheffield Teaching Hospitals NHS Foundation Trust, Glossop Road, Sheffield, S10 2JF, UK

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
