## [Decision Letter · Decision Letter 0]

27 Nov 2023

PONE-D-23-28268Sources of diagnostic delay for people with Crohn’s disease and ulcerative colitis: qualitative research studyPLOS ONE

Dear Dr. Ezaydi,

Thank you for submitting your manuscript to PLOS ONE. After careful consideration, we feel that it has merit but does not fully meet PLOS ONE’s publication criteria as it currently stands. Therefore, we invite you to submit a revised version of the manuscript that addresses the points raised during the review process.

We look forward to receiving your revised manuscript.

Kind regards,

Valérie Pittet, PhD

Academic Editor

PLOS ONE

4. Please amend your authorship list in your manuscript file to include author Naseeb Ezaydi.

Reviewers' comments:

Reviewer's Responses to Questions

**Comments to the Author**

1. Is the manuscript technically sound, and do the data support the conclusions?

Reviewer #1: Partly

Reviewer #2: Yes

2. Has the statistical analysis been performed appropriately and rigorously? 

Reviewer #1: Yes

Reviewer #2: I Don't Know

3. Have the authors made all data underlying the findings in their manuscript fully available?

Reviewer #1: Yes

Reviewer #2: Yes

4. Is the manuscript presented in an intelligible fashion and written in standard English?

Reviewer #1: Yes

Reviewer #2: Yes

5. Review Comments to the Author

Reviewer #1: In the paper Sources of diagnostic delay for people with CD nd UC: qualitative research study, a framework analysis of semi-structured interviews with 20 adult IBD patients is presented, aiming to delineate sources of diagnostic delay. Both patient’s and doctor’s DD have been addressed.

Strengths:

• Extensive ‘open’ interviews to identify potential sources of delay in the patient journey (=narrative synthesis)

• Addressing current novelties (or, in other opinion, threads) of medicine, including cure-referral, waiting lists, digitalization, language barriers and triage

Weaknesses:

• Specificity (and sensitivity) as patients with similar complaints but not having IBD were not included as comparator

• ‘Test variability’ of interviewers is unclear

• Would the patient journey be smoother when applying -currently usual- a priori (less invasive, threatening) risk factor identification, such as family history of IBD and fecal calprotectin measurement? Or would this journey be at least more specific for IBD patients? Due to what reason concludes the author that awareness of IBD would be a relevant factor, or knowledge increment of clinicians, to improve the IBD patient journey?

Specific remarks:

Abstract;

The conclusion is not necessarily deductible form the presented data.

Methods:

How was diagnostic delay precisely defined?

Were specific pretest applied or excluded in the patients journey, e.g. fecal calprotectin assessment?

Did telephone interviewing as compared to Google Meeting lead to different outcomes/sources ? Why was the cohort approached by two different communication means?

Results:

Expression of discomfort was present in 7 out of 20 participants ? Or did the 7 not expressing comfort were neutral? Anyhow, how is this number as compared to non-IBD ‘abdominal’ patients?

Line 305-312 seem to be more explanatory than rough data presentation. Why in the results section? Are other explanation possible or even likely? What about 25% of participants denying tests or investigations (line 314); representative for the health care seekers?

Line 347; is fecal testing unfamiliar/uncommon in the Sheffield area?

Discussion:

The Safer-Andersen and Pedersen models are relevant for insiders, but a short introduction night be helpful, as holds true for psychophysiological comparison. It might be better to address why these models have been applied.

Line 389 Is this a cognitive, a conative or an affective interpretation of the participant’s interview?

Reviewer #2: The topic of this manuscript is of clincal importance and sounds very intersting. Because it is not the field of my expertice, I only have a fewer minor comments: is the dignostic delay in IBD statistically signficant?

6. PLOS authors have the option to publish the peer review history of their article (what does this mean?). If published, this will include your full peer review and any attached files.

Reviewer #1: **Yes: **AA van Bodegraven

Reviewer #2: No

---

## [Author Response · Author response to Decision Letter 0]

22 Dec 2023

Thank you for the helpful and constructive comments regarding the resubmission of this manuscript. We have incorporated the feedback into the manuscript and feel that the paper is much improved as a result. We will now take each comment in turn and demonstrate how the concerns have been met on a case-by-case basis.

The formatting has been revised according to the style requirements.

We have not shared de-identified interview transcripts, as this is concordant with University of Sheffield processes for trial data sets. Furthermore, our participant consent forms and information sheets (which have been approved by Health and Care Research Support Centre, Wales Research Ethics Committee 3) stated that the transcripts would be stored securely for 10 years at the University of Sheffield, and only shared in an anonymised form if considered appropriate by the research team. Therefore, uploading interviews transcripts onto a publicly accessed data repository would oppose what participants agreed to in the informed consent procedure.

Data requests may be sent to ctru@sheffield.ac.uk.

3. Please amend your authorship list in your manuscript file to include author Naseeb Ezaydi.

This manuscript is being submitted under a group authorship ‘AWARE-IBD Diagnostic Delay Working Group’. The members of this group are listed in the acknowledgements section including Naseeb Ezaydi. 

The methods section has been amended to include the full name of the ethics committee who approved the study and the verbal consent procedure. As shown below:

“The study received ethical approval from the Health and Care Research Support Centre, Wales Research Ethics Committee 3 (16/08/2021). Informed consent was obtained verbally before all interviews. Consent was recorded onto an encrypted dictaphone and also recorded onto a paper consent form, a copy of which was sent to the participant.” Lines 157-161.

References 88 – 93 have been removed from the reference list as they are not cited in the main text. 

Reviewer 1 comments: 

Specific remarks:

Methods:

How was diagnostic delay precisely defined?

We have added the following to line 104: “For the purpose of this study, our funder defined diagnostic delay to have occurred when a person’s diagnosis by a secondary care specialist took longer than 12 months from first onset of symptoms.”

Were specific pretest applied or excluded in the patients journey, e.g. fecal calprotectin assessment?

This was a naturalistic study in which we asked participants what had happened to them. There was no attempt to prospectively manipulate pathways. Some participants will have received faecal calprotectin assessment; other will not. It was this variation in practice that the study was designed to detect.

Did telephone interviewing as compared to Google Meeting lead to different outcomes/sources ? Why was the cohort approached by two different communication means?

We have added the following at line 165: “To mitigate barriers to inclusion and participation, participants were given a choice of semi-structured interviews conducted by telephone or Google Meet...”

We did not identify any difference in the richness of interviews or the inferences we drew from them that could be related to the medium of their conduct.

Results:

Expression of discomfort was present in 7 out of 20 participants ? Or did the 7 not expressing comfort were neutral? Anyhow, how is this number as compared to non-IBD ‘abdominal’ patients?

We are struggling to respond to this comment. Only one variant on the word, “discomfort” appears in our manuscript, in Table 2, and not in connection with a frequency count. We have checked all instances of frequency counts of n=7 and n=13 and none of them appear to be to do phenomena that are synonymous with discomfort. We do not understand the other two questions. 

Line 305-312 seem to be more explanatory than rough data presentation. Why in the results section? Are other explanation possible or even likely? What about 25% of participants denying tests or investigations (line 314); representative for the health care seekers?

Sometimes in qualitative research it is useful, and accepted practice, to triangulate to the published literature during the Results. However, we realise this is unsettling to some reviewers. In deference to the reviewer, we have moved these lines to Line 431 of the Discussion.

Line 347; is fecal testing unfamiliar/uncommon in the Sheffield area?

In the journal’s PDF version of our original submitted manuscript, Line 347 reads. “347 inadequate pain control, or translation problems. Four discussed being told or”. We are struggling to understand the context of the reviewer’s question. Faecal testing is relatively common and familiar in the Sheffield area. However, this paper seeks to show variation in practice, and unevenness in its application.

Discussion:

The Safer-Andersen and Pedersen models are relevant for insiders, but a short introduction night be helpful, as holds true for psychophysiological comparison. It might be better to address why these models have been applied.

We have inserted the following, where these models are first introduced: “Psychophysiological Comparison Theory predicts that, when people notice symptoms, they try to understand their cause. They mentally compare the symptoms to their existing knowledge about illnesses. However, people are more likely to favour explanations that are less threatening. People can also describe their symptoms inaccurately because they have different perspectives and viewpoints when translating their subjective experiences into words. Social influences, especially wanting to meet certain expectations, can further change how people describe their symptoms. The Diagnostic Triage Model shows how people journey from first noticing symptoms to getting a diagnosis. There are decision points along the way - by you, your regular doctor, and specialist doctors. At each point, judgments are made – by people with symptoms, by generalists, by specialists, about how urgently the person need care. The model maps out the ups and downs on the path to diagnosis. Symptoms, people’s instincts, who they talk to, and interactions with doctors all impact the speed of progress. Delays happen when concerns are minimised; faster diagnosis comes through appropriate worry and referrals. The model aims to explain why some diagnosis journeys are smooth and some frustratingly prolonged. We chose these theories because they have evidential and coherential virtues[30]: they have shown good fit with the empirical evidence across a range of settings; and, they posit factors that plausibly cause the effect (diagnostic delay) in question which are coordinated in an intuitively plausible whole. It is appropriate to combine two theories or models to overcome the inherent limitation of each[31] - Psychophysiological Comparison Theory focusing on the patient and the Diagnostic Treatment Model on the wider system.”

Line 389 Is this a cognitive, a conative or an affective interpretation of the participant’s interview?

Line 389 in the journal’s PDF reads: "389 Change in the diagnosis prototype was changed by individuals after research - particularly on-line." The sentence appears in a discussion section and – as the use of the plural shows – relates to a generalisation based on several interviews, not that of a single participant. Our understanding is that this cognitive-conative-affective triad relates to German faculty psychology and the association psychologist, and that it is less in use since the early Twentieth Century (Hildard ER, Journal of the History of the Behavioral Sciences 16 (1980):107-117). We do not understand Psychophysiological Comparison Theory as consistent with the triad – it is a modern cognitive-affective theory, with no explicit reference to conation, and much more emphasis on cognition. For these reasons we do not feel comfortable applying synoptic/categorical theories to our data in the way the reviewer requests.

---

## [Decision Letter · Decision Letter 1]

12 Feb 2024

PONE-D-23-28268R1Sources of diagnostic delay for people with Crohn’s disease and ulcerative colitis: qualitative research studyPLOS ONE

Dear Dr. Ezaydi,

Thank you for submitting your manuscript to PLOS ONE. After careful consideration, we feel that it has merit but does not fully meet PLOS ONE’s publication criteria as it currently stands. Therefore, we invite you to submit a revised version of the manuscript that addresses the points raised during the review process.

We look forward to receiving your revised manuscript.

Kind regards,

Valérie Pittet, PhD

Academic Editor

PLOS ONE

Journal Requirements:

Reviewers' comments:

Reviewer's Responses to Questions

**Comments to the Author**

1. If the authors have adequately addressed your comments raised in a previous round of review and you feel that this manuscript is now acceptable for publication, you may indicate that here to bypass the “Comments to the Author” section, enter your conflict of interest statement in the “Confidential to Editor” section, and submit your "Accept" recommendation.

Reviewer #1: (No Response)

2. Is the manuscript technically sound, and do the data support the conclusions?

Reviewer #1: Yes

3. Has the statistical analysis been performed appropriately and rigorously? 

Reviewer #1: Yes

4. Have the authors made all data underlying the findings in their manuscript fully available?

Reviewer #1: Yes

5. Is the manuscript presented in an intelligible fashion and written in standard English?

Reviewer #1: Yes

6. Review Comments to the Author

Reviewer #1: The raised questions have been adequately addressed to improve the value of the paper for IBD care providers, such as I. Particularly, the elaboration of the used test methods is very clear. Although I gradually understand to be old-fashioned ( I plea guilty) and Germanlike (I plea not guilty) with part of my questions (19th century German theory), and to be rather anally fixated to usually put interpretation of findings in the discussion section (indeed a rather German, even Habsburgian concept, reflecting on mishappens with my parents, and as such not a major issue ;-).

I read the paper with great interest and joy, whilst reflecting on my own IBD-practice.

My apologies for unclear comments.

The authors question what was meant by "Expression of discomfort was present in 7 out of 20 participants?.....non-IBD 'abdominal' patients? ".

The questions were mentioned to address line 295. 'Thirteen patients expressed comfort in discussing symptoms', leaving potentially 13 with discomfort. Or had they/these 13 no particular mention of any feeling? The third question was asked to clarify whether this 'discussion on symptoms' was (felt to be) socially embarrassing (in general whilst having abdominal complaints or was it a specific feature of IBD patients only?).

The second enigmatic question concerned fecal testing (apparently I made a typo with the correct number of the particular line). In my country GP-initiated (laboratory) diagnostic work-up of abdominal complaints includes fecal calprotectine assessment, being an important step for referring for colonoscopy at short notice (or a red flag for rapid / prioritized specialist consultation). In Table 3 only 6 check marks are indicated for 'Elevated calprotectin', making it difficult to interpret its meaning in standard Welsh (Sheffield) practice. Hopefully, with this, the context of these particular questions is clarified.

7. PLOS authors have the option to publish the peer review history of their article (what does this mean?). If published, this will include your full peer review and any attached files.

Reviewer #1: **Yes: **AA van Bodegraven

---

## [Author Response · Author response to Decision Letter 1]

1 Mar 2024

Thank you for the helpful and constructive comments regarding the resubmission of this manuscript. We have responded to the reviewer comments below: 

1. The authors question what was meant by "Expression of discomfort was present in 7 out of 20 participants?.....non-IBD 'abdominal' patients? ".The questions were mentioned to address line 295. 'Thirteen patients expressed comfort in discussing symptoms', leaving potentially 13 with discomfort. Or had they/these 13 no particular mention of any feeling? The third question was asked to clarify whether this 'discussion on symptoms' was (felt to be) socially embarrassing (in general whilst having abdominal complaints or was it a specific feature of IBD patients only?).

We have added the following sentence to clarify expressions of comfort and discomfort amongst the twenty participants – lines 284 – 286. “Of the remaining seven participants, six expressed some discomfort in discussing symptoms during their consultations and one could not recall how they felt.”

Although we want to clarify that these were qualitative interviews involving open questions, so participants were not restricted to comfort or discomfort as responses. 

Regarding whether embarrassment during discussion of symptoms is a specific feature of IBD patients only, all twenty participants had some form of Crohn’s disease or ulcerative colitis. Interview questions focused on participant’s experiences discussing symptoms before their diagnosis, which often took the form of abdominal complaints. Therefore, we cannot say whether embarrassment when discussing symptoms is a specific feature of IBD patients only. 

2. The second enigmatic question concerned fecal testing (apparently I made a typo with the correct number of the particular line). In my country GP-initiated (laboratory) diagnostic work-up of abdominal complaints includes fecal calprotectine assessment, being an important step for referring for colonoscopy at short notice (or a red flag for rapid / prioritized specialist consultation). In Table 3 only 6 check marks are indicated for 'Elevated calprotectin', making it difficult to interpret its meaning in standard Welsh (Sheffield) practice. Hopefully, with this, the context of these particular questions is clarified.

Calprotectin is now a part of the routine work-up of suspected inflammatory bowel disease with general practitioners following local specialist guidance. This paper seeks to show patient reported variation in practice, and unevenness in its application. Clinicians in our study did not have full access to GP notes. 

Journal requirement:

References list and citations have been corrected.

---

## [Editor Report · Decision Letter 2]

20 Mar 2024

Sources of diagnostic delay for people with Crohn’s disease and ulcerative colitis: qualitative research study

PONE-D-23-28268R2

Dear Dr. Ezaydi,

We’re pleased to inform you that your manuscript has been judged scientifically suitable for publication and will be formally accepted for publication once it meets all outstanding technical requirements.

Kind regards,

Valérie Pittet, PhD

Academic Editor

PLOS ONE
---

## [Editor Report · Acceptance letter]

13 May 2024

PONE-D-23-28268R2 

PLOS ONE

Dear Dr. Ezaydi, 

I'm pleased to inform you that your manuscript has been deemed suitable for publication in PLOS ONE. Congratulations! Your manuscript is now being handed over to our production team.

Kind regards, 

on behalf of

PD Dr. Valérie Pittet 

Academic Editor

PLOS ONE